# Global Graph Counterfactual Explanation: A Subgraph Mapping Approach

**Yinhan He**                                                              *nee7ne@virginia.edu*
*Department of Electrical and Computer Engineering*
*University of Virginia*

**Wendy Zheng**                                                           *ncd9cf@virginia.edu*
*Department of Computer Science*
*University of Virginia*

**Yaochen Zhu**                                                           *uqp4qh@virginia.edu*
*Department of Electrical and Computer Engineering*
*University of Virginia*

**Jing Ma**                                                                 *jing.ma5@case.edu*
*Department of Computer and Data Sciences*
*Case Western Reserve University*

**Saumitra Mishra**                                              *saumitra.mishra@jpmorgan.com*
*JP Morgan Chase & Co.*

**Natraj Raman**                                                  *natraj.raman@jpmorgan.com*
*JP Morgan Chase & Co.*

**Ninghao Liu**                                                        *ninghao.liu@uga.edu*
*School of Computing*
*University of Georgia*

**Jundong Li**                                                             *jl6qk@virginia.edu*
*Department of Electrical and Computer Engineering*
*University of Virginia*

**Reviewed on OpenReview:** *https://openreview.net/forum?id=KQzJYI6eo0*

## Abstract

Graph Neural Networks (GNNs) have been widely deployed in various real-world applications. However, most GNNs are black-box models that lack explanations. One strategy to explain GNNs is through counterfactual explanation, which aims to find minimum perturbations on input graphs that change the GNN predictions. Existing works on GNN counterfactual explanations primarily concentrate on the local-level perspective (i.e., generating counterfactuals for each individual graph), which suffers from information overload and lacks insights into the broader cross-graph relationships. To address such issues, we propose GlobalGCE, a novel global-level graph counterfactual explanation method. GlobalGCE aims to identify a collection of subgraph mapping rules as counterfactual explanations for the target GNN. According to these rules, substituting certain significant subgraphs with their counterfactual subgraphs will change the GNN prediction to the desired class for most graphs (i.e., maximum coverage). Methodologically, we design a significant subgraph generator and a counterfactual subgraph autoencoder in our GlobalGCE, where the subgraphs and the rules can be effectively generated. Extensive experiments demonstrate the

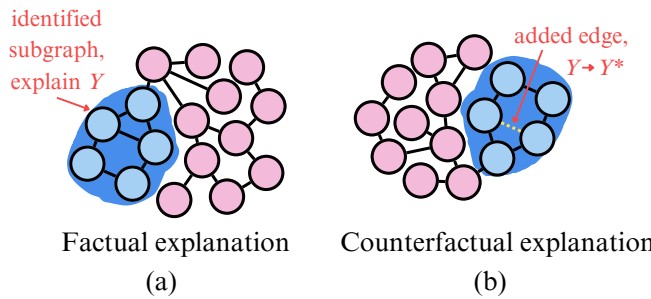

Figure 1: An illustration of GFE and GCE. (a) *GFE*. The graph is classified in the desired class because it contains a house motif, highlighted in blue. (b) *GCE*. Initially, the graph does not have a house motif and is classified as undesired. To minimally perturb the graph into the desired class, an edge (in yellow) is added, creating a house motif. The modified graph with the added edge is the counterfactual of the original graph.

superiority of our GlobalGCE compared to existing baselines. Our code can be found at `https://github.com/YinhanHe123/GlobalGCE`.

# 1 Introduction

Graph neural networks (GNNs) have achieved great success in many real-world applications, such as drug discovery (Jiang et al., 2021), financial analysis (Wang et al., 2021), and epidemic forecasting (Panagopoulos et al., 2021). For example, GNN is applied in graph classification tasks to predict whether a drug would possess the desired therapeutic efficacy (Jiang et al., 2021). The wide deployment of GNNs in high-stake applications demands the explanations of their predictions (Prado-Romero et al., 2022). Typically, GNN explainers aim to answer the following two questions: (1) What are the components (e.g., a subset of graph nodes, edges, node features, and edge features) that contribute the most to the GNN predictions? and (2) What is the *smallest perturbation* that can be applied to for the input graph to change its prediction result from an undesired class to the desired class by the GNN?

GNN explanations that focus on the first question are called *graph factual explanation* (GFE) (Tan et al., 2022), while those that focus on the second question are called *graph counterfactual explanations* (GCE) (Abrate and Bonchi, 2021; Bajaj et al., 2021; Huang et al., 2023; Lucic et al., 2022; Prado-Romero et al., 2022). In Fig. 1, we give an example of GFE and GCE for GNNs. For simplicity, we assume that the GNN to be explained classifies input graphs into the desired and undesired classes. In this example, a graph is classified as desired if it contains a house motif and undesired if it does not. In Fig. 1 (a), the smallest graph component that contributes the most to the GNN prediction is the highlighted house motif, so it serves as the factual explanation of the GNN prediction results for the input graph. In Fig. 1 (b), the original graph does not have a house motif, so the graph is classified as undesired. To minimally perturb it into the desired class, an edge (in yellow) is added to create a house motif. Here, the modified graph (the original graph with the added yellow edge) is defined as the *counterfactual explanation* of the original graph (a.k.a. 'counterfactual graph' or 'counterfactual') in the existing literature (Prado-Romero et al., 2022).

Currently, most GNN explainers only function at the local-level GCE, generating individual counterfactuals for each input graph. Despite their effectiveness, local-level GCEs face severe limitations. The most significant one is their inability to produce representative and concise *global counterfactual rules*. Here, global counterfactual rules refer to high-level graph perturbation strategies that apply to the largest proportion of graphs in an input graph domain. For example, a global counterfactual rule might suggest: "*Modifying the non-drug molecules in the TUD-AIDS (Riesen and Bunke, 2008) dataset by converting ketones and ethers into aldehydes will result **in the majority of these molecules** being classified as AIDS drugs.*" Such high-level explanations could effectively summarize model decision-making patterns and guide potential intervention directions. However, local CEs generate counterfactuals for each graph, resulting in a considerable volume of highly individualized explanations for real-world datasets (ranging from thousands to millions of graphs). Therefore, the explanations are impractical for humans to identify the global counterfactual rules.

In this paper, we aim to address the problem of generating global GCEs. Recently, Huang et al. (2023) propose *GCFExplainer*, the only known global GNN counterfactual explainer, which aims to find *a small set of counterfactual graphs*, typically no more than 100 graphs, as global GCE for GNNs. However, finding a small set of counterfactual graphs as GCE has two notable limitations: (1) ***Lack of straightforward global perturbation guidance.*** The counterfactual graph set given by GCFExplainer is still not straightforward for humans to discover general perturbation rules (e.g., where to perturb, and how to perturb) to achieve desired predictions. These perturbation rules, however, are exactly the key answers for the pivotal "what changes can be made" question of counterfactual explanation. As shown in Fig. 2, GCFExplainer still needs additional human effort to derive such answers by mapping and comparing the original graphs and counterfactual graphs, while our method can directly extract global rules as perturbation guidance. (2) ***Redundant information for explanations.*** Although GCFExplainer defines global GNN explanations as a small set of counterfactuals, each counterfactual is a whole graph that may contain redundant information irrelevant to classification. In real-world graphs, GNN predictions are often only strongly influenced by certain "significant subgraphs", such as functional groups in molecule graphs, so editing these significant subgraphs can greatly impact the classification outcomes. Therefore, the perturbation rules should focus on modifying these subgraphs for global GCE to generate counterfactuals effectively.

To address the above issues, we propose **GlobalGCE** (**Global G**raph **C**ounterfactual **E**xplanation by subgraph mapping). Given a set of input graphs, we aim to counterfactually explain a trained GNN model with human-understandable rules, each involving the map of a factual significant subgraph to a counterfactual subgraph. Specifically, the rules are learned such that by replacing the identified significant subgraph with the counterfactual subgraph according to the mapping, the explainee GNN will change the predicted class of as many of the graphs (i.e., maximum coverage constraint) in the representative dataset that contains the subgraph to another class (i.e., counterfactual constraint) as possible. However, the optimization is intractable due to the combinatorial complexity of the possible significant subgraphs and their combinations. To solve it practically, we show that the optimization target function is monotonic submodular and relaxes the problem by a greedy approximation. Specifically, we propose a novel significant subgraph generator and a counterfactual subgraph variational autoencoder, where the subgraphs and the rules can be effectively generated. This type of explanation our method provides is notably more concise and intuitive than existing efforts. We summarize our contributions as follows:

- **Problem Formulation:** We formulate a new global GNN counterfactual explanation problem based on subgraph mappings. Compared to existing works, ours enables compact global insights.

- **Novel Evaluation Metric:** We design a novel evaluation metric termed comprehensibility to measure the compactness of the graph edits produced by a GCE method, thus informing the viability of human understanding the counterfactual generation process.

- **Novel Framework:** We introduce GlobalGCE, a novel framework for global GNN counterfactual explanations providing GCE in form of CSMs. The framework is based on a carefully designed significant subgraph generator and a subgraph counterfactual autoencoder.

- **Extensive Experiments:** We perform extensive experiments on five real-world graphs to evaluate our proposed method. The results show the effectiveness of our GlobalGCE compared to other state-of-the-art baselines across multiple evaluation metrics.

## 2 Problem Definition and Evaluation Metrics

In this section, we first define the coverage of a CSM set $\mathcal{C} = \{g_i \to g_i^{cf}\}_{i=1}^k$ for $\mathcal{G}$. Then, we formulate the global-level GCE problem using the coverage of the CSMs. We also apply another metric termed proximity to measure the average graph distance between an input graph and its counterfactual acquired by $\mathcal{C}$. Finally, we propose a novel evaluation metric called *comprehensibility*, which aims to quantify how easily a human can understand the counterfactual generation process for a given original graph.

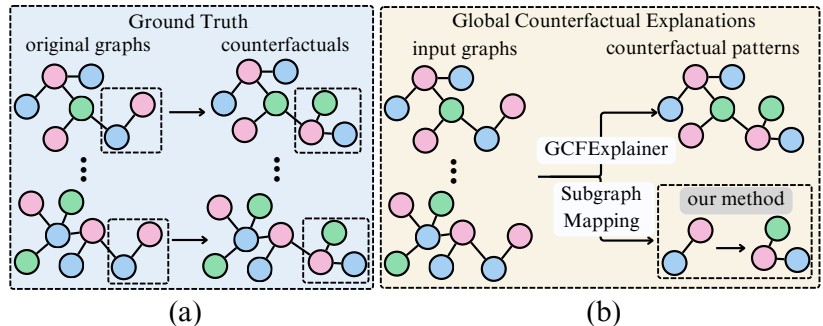

Figure 2: Comparison of global GCE provided by GCFExplainer (Huang et al., 2023) and our GlobalGCE. (a) displays ground-truth local GCEs for input graphs. In (b), the upper right shows GCFExplainer's output - a subset of counterfactuals (in this case, only one), while the lower right depicts a subgraph mapping rule as the global GCE generated by our approach. The global counterfactual rule involves changing subgraphs from the left box area to the right. This rule is not observable from the GCFExplainer output but is evident in our approach that generates subgraph mapping.

## 2.1 Global Counterfactual Explanation for GNNs

**Definition 1.** *(Coverage of a CSM Set) Given a input graph dataset $\mathcal{G}$ and a CSM set $\mathcal{C} = \{g_i \rightarrow g_i^{cf}\}_{i=1}^k$ ($\{g_i\}_{\{i=1\}}^k$ are non-isomorphic), a graph $G \in \mathcal{G}$ is 'covered' by $\mathcal{C}$ if a valid counterfactual $G^{cf}$ can be produced by applying $\mathcal{C}$. The 'coverage' of $\mathcal{C}$, denoted as **coverage**$(\mathcal{C})$, is the proportion of those covered graphs in $\mathcal{G}$.*

Here, a valid counterfactual $G^{cf}$ represents a graph that is classified in the desired class by explainee GNN. Besides, applying a CSM $\{g_i \rightarrow g_i^{cf}\}$ on graph $G$ means replacing $g_i$ in $G$ with $g_i^{cf}$ at least once. Applying a CSM set $\mathcal{C}$ on $G$ can be a combination of applications of any CSMs in $\mathcal{C}$ on $G$, including the *same CSM applied to multiple positions* of $G$ and *multiple CSMs applied simultaneously* to $G$. However, if two significant subgraphs in two CSMs overlap their positions in $G$, then the two CSMs cannot be applied simultaneously in the same modification process.

**Problem 1.** *(Global Counterfactual Explanation for GNNs) Given a GNN $\phi$ that classifies n input graphs to the undesired class 0, a CSM budget $0 < k \ll n$, the goal is to find the CSM set $\mathcal{C}$ maximizing the coverage:*

$$\max_{\mathcal{C}} \quad \textbf{coverage}(\mathcal{C})$$
$$s.t. \quad |\mathcal{C}| = k. \tag{1}$$

We can observe that this problem is NP-hard (see proof in Appendix A). Therefore, solving the exact optimization problem can be challenging. However, it is straightforward to see that the coverage function **coverage**$(\mathcal{C})$ is monotonic submodular (proof omitted for brevity), we can optimize coverage greedily while maintaining a $(1 - \frac{1}{e})$ performance lower bound (Nemhauser et al., 1978).

## 2.2 Evaluation Metrics for Global-level GCE

As suggested by the Section 2.1, we may evaluate the quality of the CSM mapping set $\mathcal{C}$ generated by a global-level GCE method by its "coverage." Besides, to evaluate a global counterfactual explainer, we also introduce proximity

$$\textbf{prox.}(\mathcal{C}) = \Sigma_{G \in \mathcal{G}_{\mathcal{C}}} \min_{\mathcal{C}} d(G, G^{cf}) / |\mathcal{G}_{\mathcal{C}}|, \tag{2}$$

where $\mathcal{G}_{\mathcal{C}}$ is the subset of $\mathcal{G}$ that is covered by $\mathcal{C}$ and the $d(G, G^{cf})$ is the $L_2$ norm of the two graphs' adjacency matrices and node/edge feature matrices. A smaller **prox.**$(\mathcal{C})$ indicates fewer adjustments to produce counterfactuals, thus making $\mathcal{C}$ more desirable global-level GCE. Finally, we design comprehensibility

$$\textbf{comp.}(\textbf{C}) = [(\Sigma_{G \in \mathcal{G}_{\mathcal{C}}} CC(G \Delta G^{cf})) / |\mathcal{G}_{\mathcal{C}}| - 0.9]^{-1}, \tag{3}$$

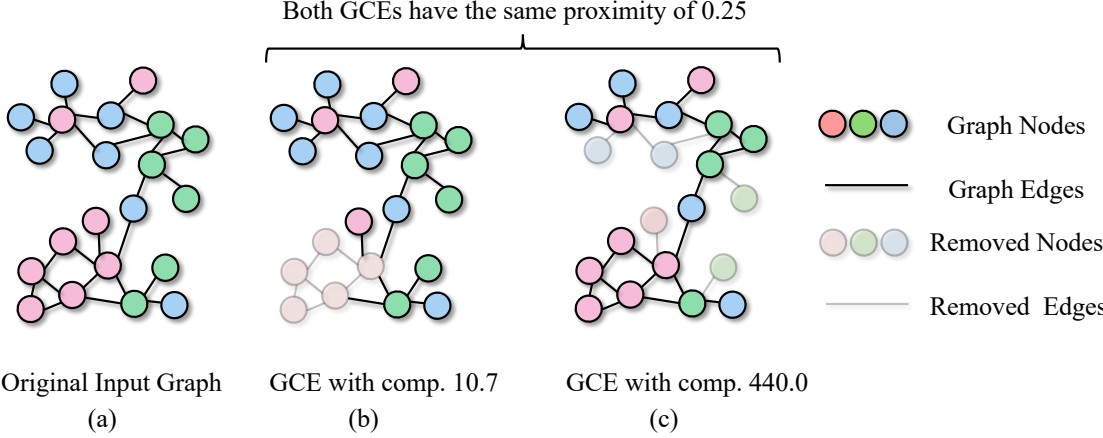

Figure 3: Illustration of the usefulness of the proposed "comprehensibility" metric. (a) shows that original input graph. (b) and (c) are two of its valid counterfactuals, they have the same proximity of 0.25, but different comprehensibility 10.7 and 440.0 respectively. (b) is apparently more human-comprehensive, which we can interpret as "removing house motif may alther the GNN prediction label for a graph." However, the graph edits for (c) is too scatter that human cannot conclude any GCE rule from the comparision of original input graph (a) and its counterfactual (c).

where $CC(\cdot)$ represents the number of connected components, and $G\Delta G^{cf}$ is the symmetric difference graph (SDG) between graphs $G$ and $G^{cf}$. The SDG consists of edges that appear in either $G$ or $G^{cf}$, but not both. The metric first calculates the average number of connected components across all SDGs between each graph $G$ in our set $\mathcal{G}_{\mathcal{C}}$ and its corresponding $G^{cf}$. Since every SDG must have at least one connected component, we subtract 0.9 (rather than 1, to avoid division by zero) to account for this baseline. Finally, we take the inverse of this value so that higher metric values indicate better comprehensibility.

## 2.3 Utility of the Proposed Comprehensibility Metric

As the proposed "comprehensibility" metric is claimed as one of the main contributions of our work, we now specifically elaborate on the intuition and usefulness of our proposed comprehensibility metric. This metric aims to measure the compactness of the graph edits during the GCE process. In fact, with a certain amount of graph edits (node/edge addition/deletion feature changes) allowed, one intuitively expects to achieve a valid counterfactual with those edits being as compact (i.e., graph edits being near-in-distance) as possible, because concentrated edits can be more easily packaged into a unified, human-comprehensible GCE rule. For example, in the Fig. 3, the graph (b), (c) are two valid counterfactuals (also called GCEs) of the original input graph (a) with the same proximity 0.25. However, graph (b) has much higher comprehensibility (440.0) than that of the graph (b), which is 10.7. We can observe that the computed comprehensibility values align with how human-comprehensive the generated GCE is: for GCE (b), we can straightforwardly conclude that "removing a house motif from the original input graph may achieve its counterfactual;" However, it is rather challenging to observe any human-comprehensible GCE rule from the comparison of input graph (a) and the counterfactual (c) since the graph edits are too scattered. A higher comprehensibility score indicates that graph edits in the counterfactual explanations are more continuous, resulting in more comprehensible GCE.

## 2.4 Adapting Metrics to Local-level GCE for Fair Comparisons

Local-level GCE (Yuan et al., 2022) methods yield a counterfactual graph for each individual input graph. Although they cannot provide a CSM set as the counterfactual explanation, we argue that our adopted three metrics are also applicable to local-GCE methods and compare the efficacy of both local-level and global-level GCE methods fairly.

Here, we first introduce how the metrics are applied in evaluating performance of local-level GCE, then we justify the fair comparison between global and local GCE methods under the metrics. The "coverage" of

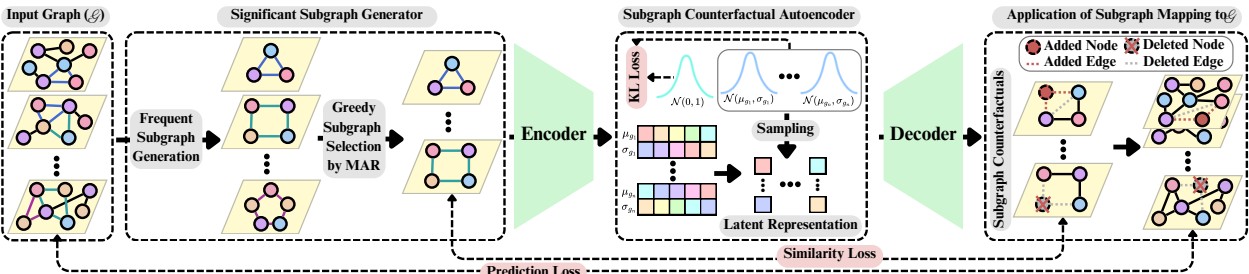

Figure 4: A overview of our GlobalGCE framework. Initially, we identify significant subgraphs from $\mathcal{G}$ based on their diversity and frequency. Then, with a counterfactual subgraph autoencoder, we generate CSMs allowing all types of graph editions. In the final step, the model selects the most representative CSMs according to their coverage.

local-level GCE methods can be computed as the proportion of the input graphs that their corresponding counterfactual generated by the GCE method is valid (i.e., the counterfactual being calssified as in the desired class by the GNN). For the "proximity" and "comprehensibility" defined in Eq. 2 and Eq. 3, we simply change the $\mathcal{G}_{\mathcal{C}}$ to the subset of the input graph set whose corresponding counterfactual provided by the certain GCE method is valid. Specifically, the "proximity" of the local-level GCE is calculated as the average graph distance between the input graphs and their counterfactual graphs. Here, the average is taken only for the input graphs whose counterfactual is valid. The "comprehensibility" measures how scattered is the graph edits from the original input graph and its corresponding counterfactual when it is valid.

The global and local GCE methods' performance can be fairly measured via the three metrics since the metrics do not directly involve the CSMs but only requires the input graphs and their corresponding counterfactual graphs. The only difference between the global-level and local-level GCE methods is that, the global-level counterfactual graphs are acquired by applying CSMs on the input graph, while the local-level graph counterfactuals are produced directly by the explainer model (typically a deep learning model) by feeding in the original input graph. The difference affects the acquisition process of graph counterfactuals, but this process is prior to and thus independent from the computation of the metrics. Therefore, the three metrics can be applied to fairly compare the performance between global and local GCE.

## 3 The Proposed Framework: GlobalGCE

Given the number of required CSMs $k$, a global GNN counterfactual explainer aim to find (output) a $k$-CSM set such that applying those CSMs to the input graph dataset achieves the highest coverage. Achieving this goal requires that (1) the subgraphs in the CSMs set be *frequent* and *diverse* enough so that each appears in a large number of different input graphs; (2) When substituting a subgraph in an input graph with its corresponding counterfactual subgraph, there should be a high probability of altering the GNN's prediction for that graph. Based on these requirements, we design our GlobalGCE with three modules (an overview of GlobalGCE is shown in Fig. 4):

- **Generation of Significant Subgraphs.** To address the first requirement, we identify frequent subgraphs in $\mathcal{G}$. We select $K(K > k)$ of them greedily based on their appearance rate (AR), i.e., the proportion of the input graphs that contain the subgraph, ensuring its diversity.

- **Counterfactual Subgraph Autoencoder (CSA).** Toward the second requirement, GlobalGCE generates a counterfactual subgraph with our designed Conditional Graph Variational Autoencoder (GVAE). Our CSA is a lightweight framework allowing all forms of graph edits.

- **Greedy Summary of Subgraph Mappings.** After the CSA generates $K$ candidate CSMs, we greedily select $k$ CSMs from the CSM candidate set by their marginal coverage in the input graph dataset. They serve as the global-level GCE of the GNN w.r.t. the input graph dataset.

### 3.1 Generation of Significant Subgraphs

Based on Problem 1, we expect to identify a significant subgraph set $\mathcal{S}$ that is *frequent*, *diverse*, and *highly influential* in GNN predictions. 'Frequent' means each subgraph in $\mathcal{S}$ appears in most input graphs in $\mathcal{G}$. 'Diverse' indicates that the subgraphs in $\mathcal{S}$ collectively cover as many different input graphs as possible. 'Highly influential' refers to the likelihood that modifying a significant subgraph will change the graph's classification outcome in the GNN. To satisfy the frequency requirement, we apply *gSpan* (Yan and Han, 2002), a classic algorithm for frequent subgraph generation, to the input graph dataset $\mathcal{G}$ with a given minimum appearance rate $\tau$ and generate a set of frequent subgraphs $\mathcal{S}$. We greedily select a set $\mathcal{S}$ from the frequent subgraphs by their appearance rate. The greedy selection guarantees the diversity of the significant subgraphs since, in each iteration, we pick the new subgraph $g$ that maximizes the number of input graphs containing it but does not contain any subgraph already selected in $\mathcal{S}$.

The last requirement is to ensure that the significant subgraphs are highly influential to the GNN prediction. However, we claim that the requirement is unnecessary in this module. This is because, during the summary stage, the subgraphs that are not influential to GNN prediction will have low coverage (see Definition 1) in the input dataset. Thus, they will be filtered out during the greedy summarization process. In the next subsection, we introduce the counterfactual subgraph autoencoder that provides the counterfactual subgraphs of those significant subgraphs.

### 3.2 Counterfactual Subgraph Autoencoder

With the identified significant subgraphs, we proceed to train the counterfactual subgraphs corresponding to those significant subgraphs utilizing a generative framework. Inspired by the application of variational autoencoder (VAE) on graphs (Simonovsky and Komodakis, 2018), we design a counterfactual subgraph autoencoder, composed of an encoder component and a decoder component, that enables fast counterfactual subgraph optimization and generalization in the input graph domain. The encoder function maps each significant subgraph $g$ onto a multi-dimensional Gaussian distribution in the latent space, denoted as $\mathcal{N}(\boldsymbol{\mu}_g, \boldsymbol{\sigma}_g * \boldsymbol{I})$, where $\boldsymbol{\mu}_g$ and $\boldsymbol{\sigma}_g$ are the mean and variance of $g$'s latent distribution. Then, we sample a latent representation $\boldsymbol{z}_g$ from this Gaussian distribution. By utilizing $\boldsymbol{z}_g$, the decoder generates $g$'s counterfactual subgraph $g^{cf}$. We optimize $g^{cf}$ by maximizing the log-likelihood that the application of the CSM $\{g \to g^{cf}\}$ successfully generates the counterfactuals of the input graphs containing $g$,

$$\ln P(g^{cf}|y^*, g) := \mathbb{E}_{G \in \mathcal{D}_g} \ln P(G^{cf}|y^*, g, G), \tag{4}$$

where $\mathcal{D}_g$ is the distribution of graphs in the input graphs distribution $\mathcal{D}$ where each graph contains the subgraph $g$, $G^{cf} := [G \setminus g] \cup g^{cf}$ is acquired by editing $g$ in $G$ to $g^{cf}$, and $y^*$ is the desired graph class. We estimate Equ. 4 utilizing the Monte Carlo approximation $\frac{1}{|\mathcal{G}_g|} \Sigma_{G \in \mathcal{G}_g} \ln P(G^{cf}|y^*, g, G)$, where $\mathcal{G}_g := \{G \in \mathcal{G} | G \supset g\}$, $\mathcal{G}$ is the input graph dataset sampled from $\mathcal{D}$. However, $\ln P(G^{cf}|y^*, g, G)$ is intractable since the posterior $\ln P(g|G^{cf}, G)$ is necessary to compute $\ln P(G^{cf}|y^*, g, G)$, but $\ln P(g|G^{cf}, G)$ is not accessible. Therefore, we utilize a viable substitute optimization target, which is a lower bound of Equ. 4 based on the Evidence Lower Bound (ELBO) (Kingma and Welling, 2013):

$$\begin{aligned} &\frac{1}{|\mathcal{G}_g|} \Sigma_{G \in \mathcal{G}_g} \ln P(G^{cf}|y^*, g, G) \\ &\geq \frac{1}{|\mathcal{G}_g|} \mathbb{E}_{Q(\boldsymbol{z}_g|g, y^*)} \ln \Pi_{G \in \mathcal{G}_g} P(G^{cf}|\boldsymbol{z}_g, y^*, G, g) - KL(Q(\boldsymbol{z}_g|g, y^*)||P(\boldsymbol{z}_g|g, y^*)), \end{aligned} \tag{5}$$

where $Q(\boldsymbol{z}_g|g, y^*)$ refers to the approximation of the posterior distribution $P(\boldsymbol{z}_g|g, y^*)$ modeled by the encoder, and $KL(\cdot||\cdot)$ means Kullback-Leibler (KL) divergence. In Equ. 5, the first term in the RHS represents the probability of the generated graph to be a counterfactual of $G$ conditioned on the latent representation $\boldsymbol{z}_g$ and the desired label $y^*$. However, due to the lack of ground-truth labels, the maximization of this term is substituted with the loss function that encourages the generation of valid counterfactual subgraphs: $-\mathbb{E}_Q[d(g, g^{cf}) + \alpha \cdot l(\phi(G^{cf}), y^*)]$, where $d(\cdot, \cdot)$ measures the distance between $g$ and $g^{cf}$, and $l(\cdot)$ is the counterfactual prediction loss, which is the negative log-likelihood between the GNN prediction $\phi(G^{cf})$ and

the desired label $y^*$. The parameter $\alpha$ controls the weight of the counterfactual prediction loss. Overall, we can write the loss function of our counterfactual subgraph autoencoder as:

$$L = \mathbb{E}_Q d(g, g^{cf}) + \alpha \cdot l(\phi(G^{cf}), y^*) + KL(Q(\boldsymbol{z}_g|g, y^*) \| P(\boldsymbol{z}_g|g, y^*)) \tag{6}$$

**Encoder.** In the encoder component, the input comprises of the node attribute matrix $\boldsymbol{X}_g$ (one-hot representation of node types), edge attribute matrix $\boldsymbol{E}_g$ (one-hot representation of edge types), as well as the graph adjacency matrix $\boldsymbol{A}_g$ of a subgraph $g$. The output is the latent representation $\boldsymbol{z}_g$ of the subgraph $g$. The encoder, modeled as a GNN, is trained to model the conditional distribution $Q(\boldsymbol{z}_g|g, y^*) = \mathcal{N}(\boldsymbol{\mu_z}(g), \mathrm{diag}(\boldsymbol{\sigma}_{\boldsymbol{z}}^2(g)))$. We minimize the KL divergence between $Q(\boldsymbol{z}_g|g, y^*)$ and the adopted prior $P(\boldsymbol{z}_g|g, y^*) = \mathcal{N}(\boldsymbol{0}, \boldsymbol{I})$ to ensure the proximity of the prior and its approximation. The latent variable $\boldsymbol{z}_g$ is sampled from the learned distribution $Q(\boldsymbol{z}_g|g, y^*)$ with the reparameterization Kingma and Welling (2013).

**Decoder.** In the decoder, the input variables are the latent representation $\boldsymbol{z}_g$ of the subgraph $g$, all graphs in $\mathcal{D}_g$, and target counterfactual label $y^*$. The output is formulated as the counterfactual subgraph $g^{cf} = (\boldsymbol{A}_{g^{cf}}, \boldsymbol{X}_{g^{cf}}, \boldsymbol{E}_{g^{cf}})$. To make our model optimizable, we adopt a probabilistic adjacency matrix, node attributes and edge attribute matrix, whose elements are continuously from the interval $[0, 1]$. We employ a similarity loss function defined as the graph distance

$$d(g, g^{cf}) = \rho d_{\boldsymbol{A}}(\boldsymbol{A}_g, \boldsymbol{A}_{g^{cf}}) + \beta \cdot d_{\boldsymbol{X}}(\boldsymbol{X}_g, \boldsymbol{X}_{g^{cf}}) + \gamma \cdot d_{\boldsymbol{E}}(\boldsymbol{E}_g, \boldsymbol{E}_{g^{cf}}). \tag{7}$$

Here, $d_{\boldsymbol{A}}, d_{\boldsymbol{X}}$ and $d_{\boldsymbol{E}}$ are $l_2$ norm of the two matrices' subtraction. After training our designed autoencoder, we discretize the adjacency, node attribute, and edge attribute matrices to generate the counterfactual subgraphs. We calculate the binary adjacency matrix $\hat{\boldsymbol{A}}_{g^{cf}}$ by thresholding these probabilistic values, setting entries to 1 if the corresponding value exceeds 0.5 and to 0 otherwise. Analogously, a one-hot node attribute matrix $\hat{\boldsymbol{X}}_{g^{cf}}$ and edge attribute matrix $\hat{\boldsymbol{E}}_{g^{cf}}$ are generated by assigning 1 to the entry with the largest value and 0 to all other entries in each row.

### 3.3 Greedy Summary of Subgraph Mappings

Now, we have generated a set of $K$ CSMs candidates $\{g_i \to g_i^{cf}\}_{i=1}^K$. Given a user-defined CSM budget $k$, we aim to select $k$ ($k \le K$) CSMs from those candidates to serve as a GCE for the largest proportion of $\mathcal{G}$. Specifically, we greedily select $k$ CSMs by the coverage function **coverage**($\mathcal{C}$). Since the coverage function is monotonic submodular [1], the selection provides a $(1 - 1/e)$ approximation. Here, we highlight three advantages of our GlobalGCE:

- **Providing Global Explanations.** We offer global-level GCE for GNNs, which differs from most existing GNN explainers focusing on generating local explanations. We are also the first model generating counterfactual subgraph mappings as GCEs.

- **Identifying Significant Subgraphs.** Compared with GCFExplainer, our model offers significant subgraphs and perturbation rules in the global-level graph counterfactual explanation, which is more informative and human-comprehensable for global explanation.

- **Facilitating Domain Knowledge Integration.** It is easy to incorporate domain constraints to GlobalGCE. For example, domain experts may rule out the counterfactual subgraphs that are infeasible to increase the model accuracy.

## 4 Experiments

In this section, we evaluate GlobalGCE with extensive experiments on five real-world datasets. Our experiments aim to answer the following research questions: **Quantative Analysis:** How does GlobalGCE perform w.r.t. the evaluation metrics compared with the state-of-the-art baselines; How do different components in GlobalGCE contribute to the performance? **Qualitative Analysis:** How does GlobalGCE

---

[1]see proof in Theorem 3 in Appendix A.

Table 1: The performance of different methods of CFEs on graphs (mean±standard deviation over five repeated executions), where "cove.", "prox." and "comp." represent coverage, proximity and comprehensibility, respectively. We also abbreviate the methods names by substituting "Explainer" with "Exp.". The best results are in bold, and the runner-up results are underlined.

| | | GNNExp. | CF-GNNExp. | CLEAR | RegExp. | GCFExp. | GlobalGCE |
|---|---|---|---|---|---|---|---|
| AIDS | cove. | 0.00±0.00 | 0.00±0.00 | 0.35±0.18 | 0.00±0.00 | 0.34±0.11 | **93.34±0.00** |
| | prox. | n/a | n/a | 15.67±2.34 | n/a | **10.03±1.17** | 16.13±0.03 |
| | comp. | n/a | n/a | 1.40±0.23 | n/a | 0.18±0.01 | **1.64±0.04** |
| NCI1 | cove. | 14.32±0.61 | 23.47±1.61 | 0.17±0.08 | 8.68±1.09 | 5.51±0.94 | **93.07±0.00** |
| | prox. | 7.09±0.05 | 8.68±0.14 | 14.28±0.00 | 11.54±0.82 | **4.21±0.33** | 15.59±0.03 |
| | comp. | 0.07±0.00 | 0.09±0.00 | **10.00±0.00** | 0.49±0.11 | 0.08±0.00 | 5.19±0.12 |
| Mutagenicity | cove. | 18.15±0.57 | 36.52±0.25 | 1.66±0.86 | 1.66±2.32 | 2.25±0.25 | **97.68±0.00** |
| | prox. | 18.25±0.25 | 15.80±0.09 | 27.28±4.78 | 7.66±4.97 | **5.22±0.36** | 25.22±0.08 |
| | comp. | 0.12±0.00 | 0.09±0.00 | 0.29±0.02 | 0.05±0.01 | 0.05±0.00 | **1.11±0.03** |
| PROTEINS | cove. | 0.00±0.00 | 0.00±0.00 | 20.47±7.98 | 0.00±0.00 | 22.79±0.93 | **72.09±0.00** |
| | prox. | n/a | n/a | 15.64±2.78 | n/a | **2.35±0.25** | 18.67±0.01 |
| | comp. | n/a | n/a | 2.62±0.48 | n/a | 0.50±0.04 | **10.00±0.00** |
| ENZYMES | cove. | 26.51±0.00 | 51.81±0.00 | 14.21±1.60 | 18.55±7.05 | 9.88±1.77 | **98.80±0.00** |
| | prox. | 12.54±0.18 | 11.64±0.13 | 18.01±1.27 | 9.33±2.62 | **3.16±0.54** | 20.87±0.00 |
| | comp. | 0.15±0.01 | 0.15±0.01 | 2.17±0.27 | 0.27±0.11 | 0.05±0.00 | **9.09±0.00** |

provide global insights from its global counterfactual explanation results? Please refer to the Appendix for supplementary experiment results.

## 4.1 Experiments Setup

*Datasets.* Our experiments utilize five real-world datasets (NCI1, Mutagenicity, AIDS, ENZYMES, PROTEINS) from TUDataset (Morris et al., 2020), where graphs represent chemical compounds with nodes as atoms and edges as bonds. They are labeled according to relevance to lung cancer, mutagenicity, HIV activity, and certain properties of proteins, respectively. See Appendix for details.

*Baselines.* We compare our GlobalGCE with the following state-of-the-art baselines. (1) **GNNExplainer** (Ying et al., 2019) is a GFE model; we apply it for GCE with some modifications on the loss function, see Appendix for details. (2) **CF-GNNExplainer** (Lucic et al., 2022) generates counterfactual ego graphs in node classification tasks. We adapt it for graph classification by taking the whole graph as input and optimizing the model using the graph label instead of the node labels. (3) **CLEAR** (Ma et al., 2022) is a generative model that produces counterfactuals of all input graphs simultaneously while preserving causality. We adapt it by removing the causality component for fair comparison. (4) **RegExplainer** (Zhang et al., 2023) generate subgraphs of input graphs as counterfactuals by graph mix-up approach and contrastive learning. (5) **GCFExplainer** (Huang et al., 2023) is the only *global-level* GCE for GNNs. This model finds counterfactuals by perturbing the original graph based on a vertex-reinforced random walk. As justified in the Section 2.4, we may apply our introduced evaluation metrics directly on those local-level GCE baselines.

*GNN Classifier.* We train a two-layer graph attention network (GAT) (Veličković et al., 2017) for graphs with edge attributes (AIDS and Mutagenicity) as a ground-truth GNN classifier. For datasets without edge attributes, we substitute the GAT with a two-layer graph convolutional network (GCN) (Kipf and Welling, 2016). Both GAT and GCN have an embedding dimension of 32, a max pooling layer, and a fully connected layer for graph classification. The model is trained with a SGD with a learning rate of 1e-3 for 500 epochs. The train/validate/test split is 50%/25%/25%. GNN accuracies are in Appendix D.1.1.

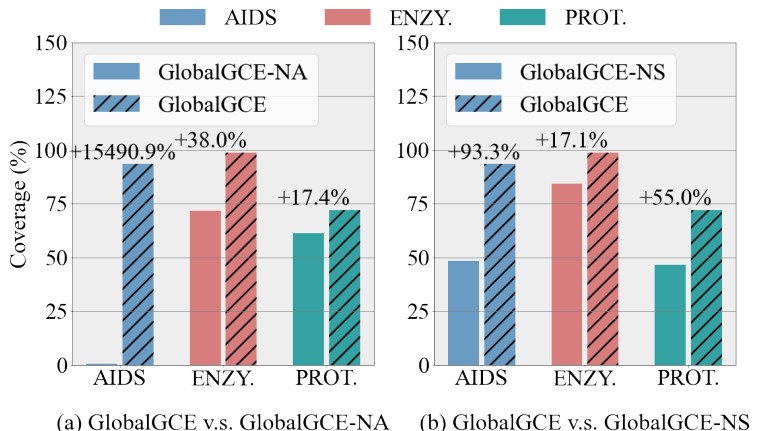

(a) GlobalGCE v.s. GlobalGCE-NA    (b) GlobalGCE v.s. GlobalGCE-NS

Figure 5: Effectiveness of the GlobalGCE's two components (shadowed bars represent the original Global-GCE framework). Here, ENZY. means ENZYMES, and PROT. means PROTEINS. We conduct experiments for GlobalGCE-NA and GlobalGCE-NS with counterfactual rules budget $k = 30$, $k = 10$, and $k = 20$ for each dataset respectively. (a) presents the coverage comparison of GlobalGCE and GlobalGCE-NA; (b) presents the coverage comparison of GlobalGCE and GlobelGCE-NS.

*Explainer Settings.* In the frequent subgraph generation, we set the minimum and maximum number of nodes to three and twenty respectively. The minimum appearance rate $\tau$ for different datasets of the frequent subgraphs is shown in the Appendix. For the counterfactual subgraph autoencoder, we allow at most two CSMs to be applied on the same input graphs to avoid combinatorial complexity (if $m$ CSMs are applicable to one input graph, $\binom{m}{k}$ combinations need to be evaluated). We set the latent space dimension as 64 and the dropout rate for the autoencoder to be 0.5. Please refer to more hyperparameters in Appendix D.1.2.

## 4.2 Quantitative Analysis

We compared GlobalGCE's coverage, proximity, and comprehensibility against state-of-the-art baselines. Results in Table 1 show: (1) **Coverage and Proximity.** GlobalGCE achieves highest/runner-up coverage across all datasets, with significant leads in AIDS (93%) and NCI (76%). Our method achieves suboptimal results in proximity since generating a large number of counterfactuals requires more deviation from the original graph. GlobalGCE is able to give higher coverage while maintaining relatively low proximity (2) **Comprehensibility.** GlobalGCE outperforms all baselines by editing only one or two self-connected significant subgraphs, resulting in concentrated edits. In contrast, GCFExplainer, the only other global graph-level GCE method, uses a random restart mechanism (i.e., randomly reselect a node/edge to edit) that produces scattered edits, leading to lower comprehensibility.

Additionally, we design two variants of GlobalGCE to examine the effectiveness of its each component. GlobalGCE-NS and GlobalGCE-NA. For GlobalGCE-NS modifies the significant subgraph generator to obtain the least significant subgraphs. GlobalGCE-NA removes the counterfactual subgraph autoencoder, and instead randomly perturbs the input graph. Fig. 5 shows that removal of any of the two components lead to significant performance degradation with the effects being the most obvious in AIDS. Next, we conduct parameter study to test our GlobalGCE's performance under different settings of hyperparameters (i.e., CSM budgets $k$ and learning rate $l$). Specifically, we vary $k$ from $k \in \{5, 10, 15, 20, 25, 30\}$ and the learning rate from $l \in \{0.0001, 0.001, 0.01, 0.015, 0.02, 0.025\}$. According to Fig. 6, we observe from (a) that higher CSM budgets lead to better coverage and comprehensibility since they enable more varied graph edits and increase the likelihood of valid counterfactuals; and our GlobalGCE demonstrates robust performance in terms of both coverage and comprehensibility across different learning rates. Lastly, we compare the average experiment time, displayed in Table 2. At first glance, GlobalGCE does not have the best performance across all datasets. However, GNNExplainer, CF-GNNExplainer, and CLEAR are local explanation methods, which means that they only need to perform on individual graphs. Thus, it is more reasonable to compare our GlobalGCE

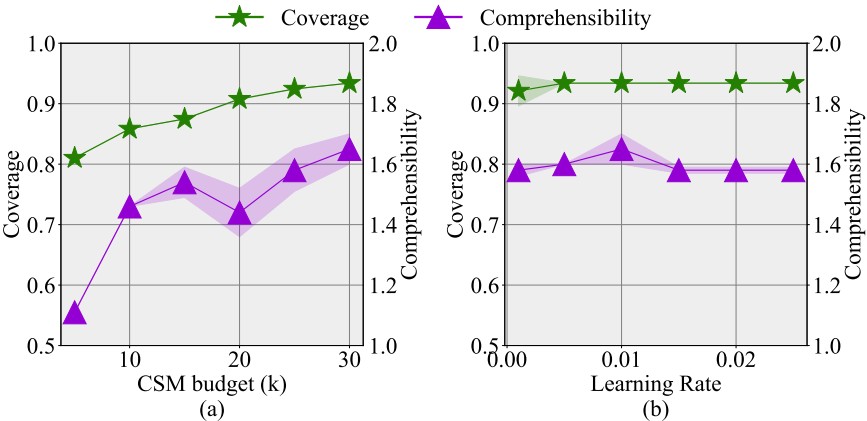

Figure 6: Parameter Sensitivity Analysis. We conduct experiments on GlobalGCE to test its coverage (in green) and comprehensibility (in purple) performance under different parameter settings on the AIDS dataset. (a) presents the GlobalGCE's performance with different numbers of counterfactual rules $k$. (b) presents GlobalGCE's performance with different $l$ under $k = 10$.

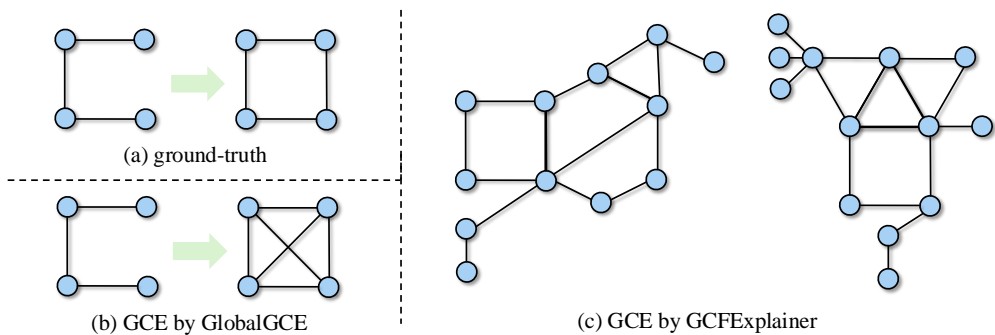

Figure 7: Case Study on A Synthetic Dataset. (a) shows the ground-truth GCE rule. (b) is the GCE in form of CSM generated by GlobalGCE. (c) displays the GCE in form of high-coverage counterfactual graphs produced by GCFExplainer, the only known gloabl-level GCE. We can observe that our GlobalGCE successfully recovers the ground-truth GCE rule with minor noise (the diagonal lines). However, one cannot uncover the ground-truth GCE from GCFExplainer due to its high noise and structual complexity.

Table 2: Average wall-clock time in seconds to run an experiment (i.e. generate counterfactual explanations) for each method-dataset combination.

|  | GNNExp. | CF-GNNExp. | CLEAR | RegExp. | GCFExp. | GlobalGCE |
|---|---|---|---|---|---|---|
| AIDS | 248.38 | 529.39 | 10.15 | 1751.03 | 497.68 | 243.20 |
| NCI1 | 53.64 | 91.69 | 3.92 | 331.54 | 160.79 | 181.77 |
| Mutagenicity | 114.54 | 205.53 | 300.84 | 839.33 | 4964.66 | 1386.20 |
| PROTEINS | 58.93 | 51.12 | 82.80 | 988.73 | 108.42 | 297.83 |
| ENZYMES | 9.23 | 20.55 | 3.93 | 57.65 | 254.21 | 130.11 |

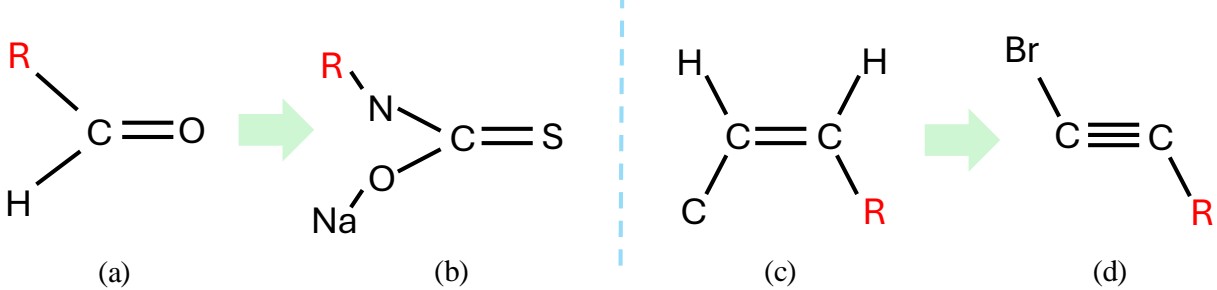

Figure 8: Case Study on Real-world Dataset. We perform GlobalGCE on AIDS and Mutagenicity with the number of rules budget $k = 10$. We show two generated CSMs: $\{(a) \to (b)\}$ for AIDS and $\{(c) \to (d)\}$ for Mutgenicity. Both CSMs reveal some chemical insights. "R" represents the rest of the compound.

to the only other global baseline GCFExplainer. As seen in Table 2, the running time for our method is comparable to that of GCFExplainer, achieving better results on AIDS, Mutagenicity, and ENZYMES.

### 4.3 Qualitative Analysis

In this subsection, we show how our GlobalGCE provides easily understandable global-level GCE on both synthetic dataset and real-world dataset [2]. First, we craft a synthetic dataset with one thousand graphs where each graph possesses nodes randomly from four to fifteen. In the graph dataset, one half of the graphs are labeled undesired (i.e., the graph does not contain any cycle), the other half are labeled desired (i.e., the graph holds at least one rectangle). The graph construction design indicates that the ground-truth GCE rule is adding one edge to enclose a four-node line to achieve a rectangle, shown in Fig. 7 (a). We conduct our GlobalGCE on the crafted dataset and found that it successfully recovers the ground-truth GCE rule (displayed in Fig. 7 (b)) with minor noise (the two diagnal lines). Meanwhile, we perform GCFExplainer, the only known global-level GCE method, on the dataset and demonstrate two top-coverage counterfactuals found by it in Fig. 7 (c). It is extremely challenging for one to extract the ground-truth GCE rule from Fig. 7 (c), proving that our GlobalGCE provides more complete global insight than GCFExplainer.

Taking AIDS and Mutagenicity (Morris et al., 2020) datasets as examples, our GlobalGCE identifies global counterfactual rules that increase the likelihood of non-AIDS drugs/non-mutagens becoming AIDS drugs/mutagens, respectively. Fig. 8 demonstrates two representative cases $(a) \to (b)$ and $(c) \to (d)$. The effectiveness of these transformations stems from several key chemical modifications: (1) **Introduction of Reactive Elements**: The addition of sulfur (=S) and bromine enhances molecular interactions, since sulfur provides antiviral properties (De Clercq and Li, 2016) and bromine forms reactive intermediates with DNA (Låg et al., 1994). (2) **Electronic Configuration Changes**: Both cases involve modifications that alter electron distributions, the sodium alkoxide group (-ONa) improves bioavailability Serajuddin (2007), while triple bonds create regions of high electron density for nucleophilic interactions (Tornøe et al., 2002). (3) **Enhanced Molecular Interactions**: Adding nitrogen (=NH) enables hydrogen bonding with target proteins (Meanwell, 2011), similar to the modified electrons interact with genetic material (DeMarini, 2004).

## 5 Related Work

### 5.1 GNN Counterfactual Explanation

There are a few studies related to GNN counterfactual explanations (Prado-Romero et al., 2023; Ying et al., 2019; Bajaj et al., 2021; Lucic et al., 2022; Ma et al., 2022; Tan et al., 2022; Huang et al., 2023). GNNExplainer (Ying et al., 2019) aims to find the counterfactual by maximizing the mutual information between the GNN's prediction and distribution of possible subgraph structures. However, it is not robust to input noise. To address this problem, RGCExplainer (Bajaj et al., 2021) generates robust counterfactuals

---

[2]Hyperparameters adjusted for best visualization

by removing edges such that the remaining graph is just out of the decision boundary. The explanation is robust because the decision boundary is in GNN's last-layer feature space, where the features are naturally robust under perturbations. Similarly, CF-GNNExplainer (Lucic et al., 2022) and $CF^2$ (Tan et al., 2022) also generate counterfactuals by removing edges. In those methods, some generated counterfactuals may violate causality, Ma et al. (2022) propose a generative model, named CLEAR, to generate causally feasible counterfactuals. Besides, there are also some methods particularly designed for a certain domain, such as biomedical and chemistry (Abrate and Bonchi, 2021; Numeroso and Bacciu, 2021; Wu et al., 2021). However, those models are all local-level GCE models, which fail to provide GCEs that humans can understand. Recently, Huang et al. (2023) propose the first global-level GCE model, GCFExplainer, which formulates global GCE as finding a small set of representative graph counterfactuals. Nonetheless, GCFExplainer is lack of straightforward global perturbation guidance and redundant information for explanations.

## 5.2 Global Counterfactual Explainer on i.i.d. Datasets

Global counterfactual explanation is attracting more attention (Rawal and Lakkaraju, 2020; Ley et al., 2022b;a; Becker et al., 2021). Among them, AReS formulates the global counterfactual explanation problem as finding a set of triples $R = \{(q_i, c_i, c_i')\}_{i=1}^n$. Here, each $(q_i, c_i, c_i')$ represents a counterfactual explanation rule: under the condition $q_i$, we can change an undesired sample to be classified as desired by changing the sample feature $c_i$ to $c_i'$. They find the counterfactual explanation rules by simultinuously optimizing their defined rules' correctness, coverage, and cost. Despite being insightful, AReS is computationally expensive because AReS selects counterfactual explanation rules from the set of all combinations of conditions ($q$s) and sample features ($c$s). To tackle this problem, Ley et al. (2022b) propose a fast rule evaluator as a preprocessor of ARes to filter out those less-likely counterfactual rules. Since using triples to represent global counterfactuals does not fit continuous sample features, GLOBE-CE (Ley et al., 2023) formulate the global counterfactual explanation as a translation of the sample features with fixed direction and varied magnitude. Specifically, for all the samples $\{\mathbf{x}_i\}_{i=1}^n$, GLOBE-CE learns a normalized vector $\delta$. The counterfactual of the samples are $\{\mathbf{x}_i + \delta k_i\}_{i=1}^n$, where $\{k_i\}_{i=1}^n$ are scalars computed by GLOBE-CE that varies among the samples. However, those models are tailored for i.i.d. datasets and non-trivial to adapt to graph datasets.

# 6 Conclusion

We study an important problem of global-level counterfactual explanations for GNNs. Specifically, we aim to find a set of CSMs as global-level GCE to achieve better global insights and to find the significant subgraphs that contribute the most to the GNN prediction. To address this problem, we propose GlobalGCE, which is composed of a significant subgraph generator, a counterfactual subgraph autoencoder, and a greedy summarization module for CSMs. To evaluate the generated counterfactuals, we design a novel and reliable comprehensibility metric. Extensive experiments validate the superior performance of our GlobalGCE.

## Acknowledgments

This work is supported in part by the National Science Foundation under grants (IIS-2006844, IIS-2144209, IIS-2223769, CNS-2154962, BCS-2228534, and CMMI-2411248) and the JP Morgan Chase Faculty Research Award.

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

## A  Proofs

**Theorem 1** The Problem 1 is NP-hard.

*Proof.* We aim to transform the problem as a maximum coverage problem, which is NP hard, thus proving that the Problem 1 is NP-hard. We assume that the node number of input graphs and their possible counterfactuals are upper limited by $L$. Then the set of all possible CSMs can be constructed as follows. For the significant subgraph set $\{g_i\}_i$ of CSMs, we build it with all subgraphs with all variations of subfeatures of the nodes and edges. Meanwhile, the counterfactual subgraph set $\{g_i^{cf}\}_i$ is composed of all possible graphs with sizes smaller than or equal to $L$. Therefore, the CSM set is the set of all graph pairs where one graph is from $\{g_i\}_i$ and the other is from $\{g_i^{cf}\}_i$ for every CSM. As graph sets $\{g_i\}_i$ and $\{g_i^{cf}\}_i$ have finite sizes, the CSM set is also finite, denoted $\{\text{CSM}_i\}_{i=1}^T$. Furthermore, we may write the set of input graphs that is covered by $\text{CSM}_i$ as $S_i$. Now, the Problem 1 is transformed to find a subset $\mathcal{S} \subset \{S_i\}_{i=1}^T$ such that $|\mathcal{S}| \leq k$ and the total number of the covered graphs is maximized. This is exactly the definition of the maximum coverage problem. $\square$

**Theorem 2** The log-likelihood of applying $\{g \to g^{cf}\}$ successfully generate the counterfactual of original input graph $G$ can be lower estimated with

$$\frac{1}{|\mathcal{G}_g|}\Sigma_{G\in\mathcal{G}_g}\ln P(G^{cf}|y^*,g,G)$$
$$\geq \frac{1}{|\mathcal{G}_g|}\mathbb{E}_{Q(\boldsymbol{z}_g|g,y^*)}\ln \Pi_{G\in\mathcal{G}_g}P(G^{cf}|\boldsymbol{z}_g,y^*,G,g) - KL(Q(\boldsymbol{z}_g|g,y^*)||P(\boldsymbol{z}_g|g,y^*)), \tag{8}$$

where the notations are consistent with those in Section 3.

*Proof.* We derive the evidence lower bound of the optimization target as follows. For a certain frequent subgraph $g$,

$$\Sigma_{G\supset g}\ln P([G \setminus g] \cup g_{cf}|Y^*,g,G)$$
$$= \ln \Pi_{G\supset g}P([G \setminus g] \cup g_{cf},|Y^*,g,G)$$
$$= \ln \Pi_{G\supset g} \int_{z_g} P([G \setminus g] \cup g_{cf}, z_g|Y^*,g,G)dz_g$$
$$= \ln \int_{z_g} Q(z_g|g,Y^*)\Pi_{G\supset g}\frac{P([G \setminus g] \cup g_{cf}, z_g|Y^*,G,g)}{Q(z_g|g,Y^*)}dz_g$$
$$\geq \int_{z_g} Q(z_g|g,Y^*)\ln \Pi_{G\supset g}\frac{P([G \setminus g] \cup g_{cf}, z_g|Y^*,G,g)}{Q(z_g|g,Y^*)}dz_g \tag{9}$$
$$= \int_{z_g} Q(z_g|g,Y^*)\Sigma_{G\supset g}\ln \frac{P([G \setminus g] \cup g_{cf}, z_g|Y^*,G,g)}{Q(z_g|g,Y^*)}dz_g$$
$$= \int_{z_g} Q(z_g|g,Y^*)\Sigma_{G\supset g}\ln \frac{P([G \setminus g] \cup g_{cf}|z_g,Y^*,G,g)P(z_g|Y^*,G,g)}{Q(z_g|g,Y^*)}dz_g.$$

For any input graph $G$ satisfying $G \supset g$, we have $P(z_g|Y^*,G,g) = P(z_g|Y^*,g)$ due to the assumption that $G$ does not effect the choice of $g_{cf}$. Hence, the Equ. 9 can be expressed as

$$\int_{z_g} Q(z_g|g,Y^*)(\Sigma_{G\supset g}\ln P([G \setminus g] \cup g_{cf}|z_g,Y^*,G,g) - |\{G|G \supset g\}|\ln \frac{Q(z_g|Y^*,g)}{P(z_g|Y^*,g)})dz_g$$
$$= \mathbb{E}_Q \ln \Pi_{G\supset g}p([G \setminus g] \cup g_{cf}|z_g,Y^*,G,g) - |\{G|G \supset g\}|KL(Q(z_g|g,Y^*)||P(z_g|g,Y^*)) \tag{10}$$

Since we do not have accurate ground-truth subgraph counterfactual $g_{cf}$, the first term can be estimated as

$$\mathbb{E}_Q d(g, g_{cf}) + \alpha \cdot l(f([G \setminus g] \cup g_{cf}), Y^*). \tag{11}$$

Therefore, the loss function is concluded as

$$L = \mathbb{E}_Q d(g, g_{cf}) + \alpha \cdot l(f([G \setminus g] \cup g_{cf}), Y^*) + KL(Q(z_g|g, Y^*)||P(z_g|g, Y^*)). \tag{12}$$

$\square$

**Theorem 3** The **coverage**($\mathcal{C}$) is monotonic submodular.

*Proof.* We will prove that coverage $\mathcal{C}$ is both monotonic and submodular. By definiton, a set function $f : 2^V \to \mathbb{R}$ is monotonic if $\forall A \subseteq B \subseteq V, f(A) \leq f(B)$, submodular if $\forall A \subset B \subset V$ and $e \in V \setminus B$, $f(A \cup e) - f(A) \geq f(B \cup e) - f(B)$.

Next, we prove each property for coverage $\mathcal{C}$: We first prove for **Monotonicity**. Let $A \subseteq B$ be two sets of CSMs. For any graph $G \in \mathcal{G}$ that is covered by $A$, $G$ must also be covered by $B$ since $B$ contains all CSMs in $A$. Therefore

$$coverage(A) = \frac{|\{G \in \mathcal{G}|G \text{ is covered by } A\}|}{|\mathcal{G}|} \leq \frac{|\{G \in \mathcal{G}|G \text{ is covered by } B\}|}{|\mathcal{G}|} = coverage(B). \tag{13}$$

Then we prove for **Submodularity**. Let $A \subseteq B$ be two sets of CSMs and $e = (g \to g^{cf})$ be a new CSM not in B. We need to prove $coverage(A \cup \{e\}) - coverage(A) \geq coverage(B \cup \{e\}) - coverage(B)$. We define $G_A$ as the set of graphs covered by $A$, $G_B$ as the set of graphs covered by $B$; and $G_e$ as the set of graphs that can be covered by applying CSM $e$. Then, the marginal gain when adding $e$ to $A$ is $coverage(A \cup \{e\}) - coverage(A) = \frac{|G_e \setminus G_A|}{|\mathcal{G}|}$. Similarly for $B$, $coverage(B \cup \{e\}) - coverage(B) = \frac{|G_e \setminus G_B|}{|\mathcal{G}|}$.

Since $A \subset B$, we have $G_A \subset G_B$, which implies $G_e \setminus G_B \subseteq G_e \setminus G_A$. Therefore $|G_e \setminus G_B| \leq |G_e \setminus G_A|$. Dividing both sides by $|\mathcal{G}|$, we have $\frac{|G_e \setminus G_B|}{|\mathcal{G}|} \leq \frac{|G_e \setminus G_A|}{|\mathcal{G}|}$. This proves that $coverage(B \cup \{e\}) - coverage(B) \leq coverage(A \cup \{e\}) - coverage(A)$. Thus, $coverage(C)$ is submodular. As we have proven both monotonicity and submodularity, we can conclude that $coverage(C)$ is monotonic submodular. $\square$

## B  Limitation

While our proposed GlobalGCE framework demonstrates significant improvements in generating global counterfactual explanations for GNNs, there are several limitations that warrant further investigation. First, the computational complexity of our approach increases with the size and number of input graphs, which may limit scalability to extremely large graph datasets. Additionally, our method currently focuses on binary classification tasks, and extending it to multi-class or regression problems requires careful consideration. The subgraph mapping rules generated by GlobalGCE, while interpretable, may not capture all nuances of the GNN's decision-making process, especially for highly complex graph structures. Furthermore, the effectiveness of our approach relies on the quality of the identified significant subgraphs, which may not always align perfectly with the most influential substructures for the GNN's predictions.

## C  Time Complexity Analysis

**Subgraph search.** We utilized gSpan Yan and Han (2002), a mature algorithm for frequent subgraph generation. The time complexity of gSpan is $O(k \times n \times v!)$, where $k$ is the average size of subgraphs explored (which is significantly limited by the minimum appearance rate $\tau$), $n$ is the number of graphs in the dataset, and $v$ is the average number of vertices in each graph. This complexity arises from the algorithm's exploration of the DFS Code Tree, where each node requires subgraph isomorphism testing (contributing the $v!$ factor), and the depth of exploration is controlled by both the graph size and the minimum appearance rate $\tau$. The $n$ factor comes from needing to check each candidate subgraph against all graphs in the dataset. A higher $\tau$ reduces $k$ by pruning more branches of the search space, as only frequent subgraphs meeting the threshold are explored. Therefore, although the time complexity can be theoretically high, we may still conduct the gSpan with adjustable $\tau$ when the graph size scale to be very large. Specifically, if gSpan does not return

result after a determined time period, we increase the minimum appearance rate, the running time will decrease polynomially/super-polynomially (refer to Fig 5 of Yan and Han (2002)).

**Counterfactual generation.** The time complexity for training the subgraph counterfactual VAE can be analyzed per training iteration. For each significant subgraph $g$, we process all graphs containing it (denoted as $|G_g|$) in batches of size b. For each graph in a batch, we perform VAE operations on the subgraph ($O(n\check{s})$ for n-node subgraphs), subgraph matching to find insertion locations ($O(Nn)$ for N-node original graphs), insert the counterfactual subgraph ($O(n^2)$), and run forward/backward passes through the GNN ($O(N^2)$). With i training iterations, the total time complexity is $O(i*|G_g|/b*(n\check{s}+Nn+N\check{s}))$. In general, as shown in Table 2, our method consumes significantly less time than the only existing global-level graph counterfactual explanation work GCFExplainer.

# D    Reproducibility

In this section, we provide more details of model implementation and experiment setup of our evaluation results.

## D.1    Details of the Model Implementation

### D.1.1    GNN as the Prediction Model

We train a two-layer graph attention network (GAT) for graphs with edge attributes (AIDS and Mutagenicity) as a ground-truth GNN classifier. For those datasets without edge attributes, we substitute the GAT with a two-layer graph convolutional network (GCN). Both the GAT and GCN have embedding dimension 32, a max pooling layer, and a fully connected layer for graph classification. The model is trained with the PyTorch Adam optimizer using dataset-specfic hyperparameters shown in Table 3.

Table 3: GNN Prediction Model hyperparameter values for each dataset.

|  | NCI1 | AIDS | Mutagenicity | PROTEINS | ENZYMES |
|---|---|---|---|---|---|
| train epochs | 100 | 300 | 100 | 400 | 400 |
| learning rate | 0.1 | 0.01 | 0.01 | 0.0001 | 0.001 |
| weight decay | 0.001 | 0.01 | 0.001 | 0.0001 | 0.01 |

The train/validate/test split is 50%/25%/25%. The accuracy of the GNNs is shown in Table. 4

Table 4: Graph classification accuracy (%) of the GNN classifiers.

|  | NCI1 | Mutagenicity | AIDS | PROTEINS | ENZYMES |
|---|---|---|---|---|---|
| Training | 91.32 | 97.62 | 99.89 | 91.00 | 99.00 |
| Validation | 95.92 | 67.24 | 99.78 | 64.00 | 72.00 |
| Testing | 68.40 | 72.84 | 99.57 | 62.00 | 64.00 |

### D.1.2    Details of GlobalGCE

Our GlobalGCE comprises three stages: generation of significant subgraphs, counterfactual subgraph autoencoder, and greedy summary of CSMs.

Table 5: Minimum appearance rate of the frequent subrgaphs in the input graph dataset. Here, min. AR means minimum appearance rate.

|  | NCI1 | AIDS | Mutagenicity | PROTEINS | ENZYMES |
|---|---|---|---|---|---|
| Min. AR | 64.10% | 20.44% | 92.59% | 28.57% | 10.20% |

- **Generation of Significant Subgraphs.** We first find frequent subgraphs, where we set the minimum and maximum number of nodes of frequent subgraphs to three and twenty, respectively. In addition, the minimum appearance rate $\tau$ for different datasets of frequent subgraphs (i.e., a subgraph is defined as frequent subrgaph if it appears in more than $\tau$ proportion of the input graphs) is shown in Table 5, which is determined by trials to ensure that the generated number of frequent subgraphs can achieve the predefined budget $K$.

- **Counterfactual Subgraph Autoencoder.** We allow at most two CSMs to be applied on the same input graphs. We set the latent space dimension as 64 and the dropout rate to 0.5. We train the model with dataset-specific hyperparameters, which is listed in Table 6.

- **Omission of Greedy Selection by Appearance Rate.** According to Section 3, the significant subgraphs are greedily selected from the frequent subgraphs with the appearance rate function; however, extensive empirical evaluations suggest that the frequent subgraphs themselves are diverse enough, and the greedy selection *does not gain higher performance* for our GlobalGCE. Therefore, in our experiments, we omit the greedy selection and identify significant subgraphs as those subgraphs with $k$ highest appearance rates.

- **Graph Distance Calculation.** We approximate graph edit distance (GED) Gao et al. (2010) with

$$d(g, g^{cf}) = \rho \cdot d_{\boldsymbol{A}}(\boldsymbol{A}_g, \boldsymbol{A}_{g^{cf}}) + \beta \cdot d_{\boldsymbol{X}}(\boldsymbol{X}_g, \boldsymbol{X}_{g^{cf}}) + \gamma \cdot d_{\boldsymbol{E}}(\boldsymbol{E}_g, \boldsymbol{E}_{g^{cf}}). \tag{14}$$

Here, $d_{\boldsymbol{A}} = ||A_1 \odot (1 - A_2)||_2$, $d_{\boldsymbol{X}}$ and $d_{\boldsymbol{E}}$ are calculated with $l_2$ pairwise distance. As the computation of $d_{\boldsymbol{A}}$ is time-consuming, we simplify this value as cross-entropy with logits in the training of the counterfactual autoencoder. In all other situations, such as the evaluation of GlobalGCE and all baselines, we adopt the definition with the dot product. We set the $\alpha = 10, \beta = \gamma = 1$ to emphasize the counterfactual's structural change.

Table 6: GlobalGCE hyperparameter values for each dataset.

|  | NCI1 | AIDS | Mutagenicity | PROTEINS | ENZYMES |
|---|---|---|---|---|---|
| train epochs | 100 | 300 | 100 | 100 | 100 |
| topk | 20 | 30 | 20 | 20 | 10 |
| learning rate | 0.01 | 0.01 | 0.1 | 0.01 | 0.01 |

## D.2 Details of Experiment Setup

### D.2.1 Baseline Settings

- **GNNExplainer**: For each graph, GNNExplainer (Ying et al., 2019) outputs an edge mask that estimates the importance of different edges in model prediction. We adapt this model to counterfactual generation taking the prediction loss term as the negative likelihood of the masked graph a being classified as desired by GNN rather than undesired. We remove edges with edge mask weights smaller than the threshold 0.5. The perturbation on node features cannot be designed as straightforwardly as the perturbation on the graph structure; thus, we do not perturb graph node features in GNNExplainer.

- **CF-GNNExplainer**: CF-GNNExplainer (Lucic et al., 2022) is originally proposed for node classification tasks, and it only focuses on the perturbations on the graph structure. Originally, for each explainee node, it takes its neighborhood subgraph as input. To apply it on graph classification tasks, we use the whole graph as the neighborhood subgraph and assign the graph label as the label for all nodes in the graph. We set the number of iterations to generate counterfactuals as 500.

- **CLEAR**: CLEAR (Ma et al., 2022) is a generative model that produces counterfactuals of all the input graphs simultaneously while preserving causality. This model generates both a graph adjacency matrix and a graph node feature matrix perturbation, which allows all forms of graph edition

such as node/edge addition/deletion/feature perturbation. We adapt it by removing the causality component for fair comparison. We train the model with 500 epochs, other hyperparameters are in default according to Ma et al. (2022).

- **RegExplainer**: RegExplainer (Zhang et al., 2023) explains graph regression models with information bottleneck theory to help solve the distribution shift problem. Although the original model is tailored for graph regression models, it can be directly applied to the graph classification tasks. We implement the RegExplainer with GNNExplainer as the base explainer model in Algorithm 2 of (Zhang et al., 2023) and train the model for 500 epochs. Other hyperparameters are tuned for the best performance in each dataset.

- **GCFExplainer**: GCFExplainer (Huang et al., 2023) is the only global-level GCE for GNN utilizing vertex reinforcement random walk. "Vertex-reinforced" means that the transition probability at each step of the walk is inversely proportional to the frequency of visits to neighboring nodes. We apply this model for each dataset with the maximum steps as 1000. Other hyperparameters are in default according to Huang et al. (2023).

### D.3  Experiment Settings

All the experiments are conducted in the following environment:

- Python==3.9

- Pytorch==1.11.0

- torch-geometric==2.1.0

- torch-scatter==2.0.9

- torch-sparse==0.6.15

- Scipy==1.9.3

- Networkx==3.0

- Numpy==1.23.4

- Cuda==11.6

- gSpan-mining==0.2.3

#### D.3.1  Datasets

We utilize five real-world datasets from TUDataset (Morris et al., 2020), for the meta-data, see Table 7.

Table 7: Metadata of the adopted real-world datasets. Here, Muta. means Mutagenicity. Av. Nodes and Av.Edges are the average number of graph nodes and edges, N.C. means the number of node class

| Dataset | Graphs | Av. Nodes | Av. Edges | N.C. |
|---|---|---|---|---|
| NCI1 | 4,110 | 29.87 | 32.30 | 10 |
| Mutagenicity | 4,308 | 30.32 | 30.77 | 10 |
| AIDS | 1,837 | 15.69 | 16.20 | 9 |
| PROTEINS | 1,113 | 39.06 | 72.82 | 3 |
| ENZYMES | 600 | 32.63 | 62.14 | 2 |

- **NCI1:** The NCI1 dataset is a collection of chemical compounds screened for their ability to inhibit the growth of a panel of human tumor cell lines. Each compound is represented as a graph, where vertices are atoms, and edges represent chemical bonds. This dataset is widely used in graph kernel methods for machine learning, especially for graph classification tasks.

- **Mutagenicity:** The Mutagenicity dataset comprises chemical compounds labeled according to their mutagenic effect on a bacterium. Similar to NCI1, each compound is graphically represented, focusing on the atom and bond structure. It serves as a key resource for studying graph machine-learning methods.

- **AIDS:** This dataset contains chemical compounds that are potential anti-HIV agents. Each graph represents a compound's molecular structure, capturing the connectivity and properties of atoms. The AIDS dataset is often used in graph-based machine learning for tasks such as compound classification and pattern recognition.

- **ENZYMES:** ENZYMES is a set of protein tertiary structures. Each protein structure is represented as a graph, where nodes are amino acids, and edges connect neighboring amino acids. This dataset is particularly useful for research in graph-based protein function prediction and structural analysis.

- **PROTEINS:** The PROTEINS dataset consists of protein structures, represented as graphs. Nodes in these graphs represent secondary structure elements, connected by edges if they are neighbors in the amino acid sequence or 3D space. This dataset facilitates the study of graph-based methods in understanding protein structure-function relationships.

