# OpenReview forum: "Global Graph Counterfactual Explanation: A Subgraph Mapping Approach"
_TMLR — Accepted by TMLR_

### Review · Reviewer_u3YN · 2024-11-12

**Summary Of Contributions:**

1. This work introduces a new problem, global GNN counterfactual explanation based on subgraph mappings.
2. The authors design a new evaluation metric termed comprehensibility to measure the compactness of graph edits produced by a GCE method.
3. The authors introduce GlobalGCE, a new framework for global GNN counterfactual explanations providing GCE in form of CSMs.

**Audience:**

Yes

**Broader Impact Concerns:**

I have no concerns.

**Claims And Evidence:**

Yes

**Requested Changes:**

1. Please refer to the weaknesses above, and address them as much as you can.
2. In Eq. (2), it seems wrong that $C$ is in the subscript of the min function.
3. The comprehensibility of graphs in Fig 3 is presented differently in text and in the caption.

**Strengths And Weaknesses:**

Strengths
1. The paper is well-organized in general. The authors define a new problem, introduce a new evaluation metric, and then propose their own approach to solve the problem.
2. It’s good that the proposed measures are not only applicable for global GCE, but also for local GCE for a fair comparison with existing methods.
3. The authors performed diverse experiments, and the proposed approach shows good empirical performance compared with recent models.

Weaknesses
1. I can understand the meaning of comprehensibility thanks to Section 2.3, but the mathematical intuition behind Eq. (3) is hard to capture. I suggest improving the representation and explanation of Eq. (3) in Section 2.2.
2. Eq. (7) seems a little too naive. Since we know that X and E are one-hot vectors, there should be a better way to measure the difference, e.g., KL divergence, or the likelihood of X_g (or E_g) given their counterfactuals.
3. It would be nice to include a proof on why coverage(C) is monotonic submodular in Appendix, since it is a core part of this work.

---

> ### Author Response · Authors · 2024-11-19
>
> We sincerely thank the reviewer u3YN for their efforts in reviewing our manuscript. We will edit our manuscript according to the reviewer's requested changes. Here, we provide point-by-point answers to address the reviewers' concerns.
> ## Weakness 1
> Thank you for this helpful feedback. Please allow us to give a more detailed explanaiton on Equ.3. To do this, we first breakdown the comprehensibility metric $comp.(C) = [(\Sigma_{G\in G_C} CC(G\Delta G^{cf}))/|G_C| - 0.9]^{-1}$:
> - $G\Delta G^{cf}$ is the symmetric difference graph (SDG) between the original graph $G$ and its counterfactual $G^{cf}$.
> - $CC(\cdot)$ counts the number of connected components in this difference graph.
> - The averaging term $(\Sigma_{G\in \mathcal G_\mathcal C} CC(G\Delta G^{cf}))/|\mathcal G_\mathcal C|$ represents the mean number of disconnected edit regions for $\mathcal{G}_\mathcal{C}$, the subset of graphs covered by $\mathcal{C}$.
> - Subtracting 0.9 (instead of 1) creates a small offset to avoid division by zero.
> - Taking the inverse (-1 power) is to make the measure is more human intuitive, as, now, higher values indicate better comprehensibility.
>
> **Rationale**:
> The metric quantifies how "compact" the edits are between $G$ and $G^{cf}$. When edits are concentrated in fewer regions (lower number of connected components in $G\Delta G^{cf}$), the denominator becomes smaller, resulting in a larger (better) comprehensibility. Conversely, scattered edits create more disconnected components, leading to a smaller final value.
>
>
> ## Weakness 2
> We appreciate the reviewer's insightful observation about the distance metrics in Equation 7. First, we need to clarify that $X$ and $A$ are not always one-hot. When they are generated by the graph decoder before processing them into one-hot, they are continuous values. Second, we present the three key reasons for our choice: (1) Empirical effectiveness: Our experiments show that the approximity loss implemented by simple $l_2$ norm achieved satisfactory GCE performance, it is not necessary to utilize more complex metrics such as KL divergence to introduce more computation costs. (2) Numerical stability: KL divergence can be unstable when distributions are close to zero (property of log computations), requiring additional smoothing techniques. (3) Interpretability: The l2 norm provides more intuitive distance measures that are easier to tune and interpret. We agree this is an interesting direction for metric improvement, we will explore more sophisticated distance metrics in future work. Thank you for bringing this to our attention.
>
> ## Weakness 3
> We will prove that coverage $\mathcal{C}$ is both monotonic and submodular. By definiton, a set function $f:2^V\rightarrow \mathbb{R}$ is:
> - Monotonic if $\forall A ⊆ B ⊆ V, f(A) ≤ f(B)$,
> - Submodular if $\forall A \subset B \subset V$ and $e \in V \setminus B$, $f(A \cup {e}) - f(A) ≥ f(B \cup {e}) - f(B)$.
>
> Next, we prove each property for coverage $\mathcal{C}$:
>
> **Monotonicity**
>
> Let $A ⊆ B$ be two sets of CSMs. For any graph $G ∈ \mathcal{G}$ that is covered by $A$, $G$ must also be covered by $B$ since $B$ contains all CSMs in $A$. Therefore
> $coverage(A) = \frac{|\{G \in \mathcal{G} | G \text{ is covered by } A\}|}{|\mathcal{G}|} \leq \frac{|\{G \in \mathcal{G} | G \text{ is covered by } B\}|}{|\mathcal{G}|} = coverage(B).$
>
> **Submodularity**
>
> Let $A ⊆ B$ be two sets of CSMs and $e = (g → g^{cf}$) be a new CSM not in B. We need to prove
> $coverage(A \cup \{e\}) - coverage(A) \geq coverage(B \cup \{e\}) - coverage(B)$. We define $G_A$ as the set of graphs covered by $A$, $G_B$ as the set of graphs covered by $B$; and $G_e$ as the set of graphs that can be covered by applying CSM $e$. Then, the marginal gain when adding $e$ to $A$ is $coverage(A \cup \{e\}) - coverage(A) = \frac{|G_e \setminus G_A|}{|\mathcal{G}|}.$ Similarly for $B$, $coverage(B \cup \{e\}) - coverage(B) = \frac{|G_e \setminus G_B|}{|\mathcal{G}|}.$
>
> Since $A \subset B$, we have $G_A \subset G_B$, which implies $G_e \setminus G_B \subseteq G_e \setminus G_A.$ Therefore $|G_e \setminus G_B| \leq |G_e \setminus G_A|$. Dividing both sides by $|\mathcal{G}|$, we have
> $\frac{|G_e \setminus G_B|}{|\mathcal{G}|} \leq \frac{|G_e \setminus G_A|}{|\mathcal{G}|}.$
> This proves that $coverage(B \cup \{e\}) - coverage(B) \leq coverage(A \cup \{e\}) - coverage(A).$ Thus, $coverage(C)$ is submodular.
>
> Since we have proven both monotonicity and submodularity, we can conclude that $coverage(C)$ is monotonic submodular.

---

### Review · Reviewer_yrGS · 2024-11-19

**Summary Of Contributions:**

This paper proposes a novel problem of global graph counterfactual explanation that is to extract a set of subgraph transformation rules that can make the most of the input graph to be classified to the target class.

For this task, the author proposes the novel three metrics: coverage calculates haw many input graphs can be transformed to the desired class by the rules. Proximity calculates the magnitude of the change of the graph. Comprehensibility calculates the number of connected subgraphs required to add or remove in the graph transition process.

Also, the author proposes a novel method for the global counterfactual extraction consists of three modules. First extracts the candidate subgraphs from input graphs considering frequency, diversity, and influence. Then trains the encoder-decoder model to transform the input subgraph to the different subgraph such that the transformed graph is classified to the desired category. Finally filters the candidate subgraph transformation rules with respect to the coverage.

The author experimentally demonstrates that the proposed method demonstrates better performance than the existing local counterfactual explanation methods in the proposed three metrics. Furthermore, the author conducts ablation studies and visualizations to demonstrates the effectiveness of the proposed method.

**Audience:**

Yes

**Broader Impact Concerns:**

The paper does not have broader impact section.

The paper has algorithmic limitation section in the appendix.

I think the potential negative impact would be small, since the contribution is the extension of the existing task.

**Claims And Evidence:**

Yes

**Requested Changes:**

Overall, my evaluation is positive at this stage even without the suggestion below.

Additional discussion about the proposed metric would clarify the importance of each metric.

Discussion about multiple transition rules case are just from my interest.

Discussion about evaluation of existing methods would help clearing the fairness of the comparison.

The explanation about distance in equation (2), problem setting and about ground-truth labels in section 3.2. would help improve the clarification.

**Strengths And Weaknesses:**

Strength

The author proposes the novel task of global graph counterfactual explanation, evaluation metrics for the proposed task, and effective method to tackle the proposed global graph counterfactual explanation task. The paper has good novelty.

The proposed evaluation metrics and proposed prediction methods are well-discussed and seem suitable for the proposed task.

The proposed method demonstrates good performance with respect to the proposed metric.

The author conducts several ablation studies and visualizations.


Weakness

As for the proposed metric, proximity and comprehensibility indirectly depend on the coverage since they are calculated only on the covered samples. Therefore, it may be difficult to compare the methods with different coverage scores using these scores.

In the bottom of page 3, the author discusses the case where we apply the subgraph transition multiple times. I find this case interesting. However, it seems the proposed model training method and evaluation does not care this case. In fact, how many test case exist where multiple application of the transition is required.

According to 2.4, is the coverage of existing local method same as a sort of success rate ignoring k? If so, I would like to see the reason of much worse performance in Table 1.
Further, can we compare the methods with existing metric?

In the equation (2), the definition of the distance d is not explained. Since there is an ambiguity, e.g. graph edit distance, number of transition rule in C applied, distance proposed in the equation (7). It would be clearer to add explanation just before or after the equation (2).

In section 3.2, the author says the ‘lack of ground-truth labels’. Is that indicate y*? I would like the clarification of the problem setting i.e. what sort of information can we access during the training of subgraph extraction rule. Further, the more discussion about justification of the replace of this term with equation (6) would be helpful for the clarification.

---

> ### Author Response · Authors · 2024-12-01
>
> We sincerely appreciate the reviewer yrGS for the time and efforts you've dedicated to reviewing and providing invaluable feedback to enhance the quality of this paper. We provide a point-to-point reply below for the mentioned concerns and questions.
> ## Weakness 1
> To eliminate this dependency, we would need to compute these two metrics across all graphs rather than just the covered ones. However, we believe this approach would introduce noise and make it more challenging to compare methods effectively. For uncovered graphs, the generated counterfactuals are predicted to belong to the same class as the original graph, suggesting that the number of edits was insufficient in altering the prediction. Consequently, the proximity for these sample-counterfactual pairs would be very low. Additionally, the comprehensibility would also be low due to the small number of edits. Including uncovered samples in the calculations of proximity and comprehensibility for methods with low coverage would artificially decrease proximity and increase comprehensibility. This could lead to a misleading conclusion that the method performs better in terms of these metrics.
>
> ## Weakness 2
>
> As the name suggests, frequent subgraphs occur commonly within the input graphs, often appearing multiple times due to their small size (typically three or four nodes). During preprocessing, we identify all occurrences of each frequent subgraph and replace them with the corresponding counterfactual subgraph during training and evaluation. We opted to replace all occurrences simultaneously over replacing only one occurance to avoid the ambiguity of selecting which instance to replace. Additionally, allowing for all possible replacement combinations would risk a combinatorial explosion in complexity.
>
> ## Weakness 3
>
> Yes, the local methods aim to generate a counterfactual for every input graph. We believe their poor performance is likely due to the additional processing applied to the counterfactuals prior to evaluation. Specifically, we discretized the node attribute matrix, edge attribute matrix, and adjacency matrix to ensure the generated counterfactual data structure aligns with the format of the input graphs. In contrast, we observe that the baseline methods do not perform this step and instead use continuous values when reporting their results.
>
> Regarding the second question, we argue that these methods can be compared using the current metrics because the metrics evaluate only the final outputs, the generated counterfactuals, making them method-agnostic. Our goal is to determine whether our global method, even with a limited budget, can match or exceed the performance of local methods. This is why we selected these three metrics.
>
> ## Weakness 4
>
> Thank you for discovering this discrepancy. The definition of 𝑑 in Equation 2 refers to the sum of the graph feature matrix distances as defined in Equation 7. Specifically, the graph edit distance 𝑑 is calculated as the sum of the 𝐿2-norms of the differences between the node attribute matrices, edge attribute matrices, and adjacency matrices of the counterfactual and the input graph.
>
> ## Weakness 5
>
>
> Before answering the question threads, we apologize for the misunderstanding. "the lack of ground-truth labels" is a typo, it should be "the lack of ground-truth counterfactuals." (1) "the lack of ground-truth counterfactuals" does not indicate $y^*$, it indicate $G^{cf}_{ground-truth}$ which is required for calculating $P(G^{cf}|y^*,g,G)$ in Equ.5. (2) We can access the input graphs' structures, their node/edge features. For any candicate graph, we can access its graph label via the ground-truth GNN. (3) During training, the training datasets are input graphs whose ground-truth counterfactual graphs are unknown, therefore the term $P(G^{cf}|y^*,g,G)$ in Equ.5 can not directly be calculated, so in order to "implicitly" optimize the $P(G^{cf}|y^*,g,G)$, we instead try to minimize the generated counterfactual's distance from the original input graph and its probability of being classified as desired.

---

### Review · Reviewer_TSmf · 2025-02-15

**Summary Of Contributions:**

This paper tackles the challenge of generating global-level counterfactual explanations for Graph Neural Networks (GNNs). Rather than producing separate edits for each individual graph, the authors propose a small set of subgraph-mapping rules that, when applied, flip many graphs’ predictions to a desired label. The framework—called GlobalGCE—operates in three stages: (1) it identifies significant subgraphs that appear frequently across the dataset, (2) uses a lightweight autoencoder to learn how to replace each significant subgraph with a counterfactual subgraph, and (3) greedily selects the best subgraph-mapping rules to maximize coverage. The paper also introduces a new comprehensibility metric. Experiments on multiple datasets demonstrate that GlobalGCE can flip a larger proportion of graphs’ labels than prior methods, often with more compact edits and clearer global insights.

**Audience:**

Yes

**Claims And Evidence:**

Yes

**Requested Changes:**

Aforementioned in weakness

**Strengths And Weaknesses:**

Strengths:
- The paper addresses a significant and timely problem within the domain of GNN explainability.
- The newly proposed comprehensibility metric is intuitive and effectively complements standard interpretability metrics such as proximity.
- The experiment results seem reasonable.

Weakness:
- Stating that “existing works on GNN counterfactual explanations primarily concentrate on the local-level perspective” is not entirely precise. Several model-level (i.e., global) explanation approaches already exist—e.g., XGNN [1], GNNBoundary [2], GDM [3], D4-Explainer [4], and Graphon-Explainer [5]. Direct comparison with these methods would help validate the effectiveness of the proposed approach.
- Additional datasets and baselines from these aforementioned works would strengthen the empirical evaluation and bolster claims about method generalizability.
- The idea of identifying common, influential motifs is not entirely new. Relevant efforts such as [6-7] also aim to discover representative motifs to explain model behavior. A comparison or discussion regarding these works would contextualize the contributions more convincingly.
- Although Figure 3 is illustrative, the assertion that comprehensibility is unequivocally “better” than other metrics (e.g., proximity) demands more rigorous justification, such as theoretical analysis or more systematic empirical motivations. In many real-world scenarios, there is no readily available “ground-truth motif” to benchmark comprehensibility. The quality of the set $\mathcal{G}$ can strongly influence comprehensibility scores, and the approach may perform differently on larger, more complex graphs.

[1] XGNN: Towards Model-Level Explanations of Graph Neural Network. KDD'20

[2] GNNBoundary: Towards explaining graph neural networks through the lens of decision boundaries. ICLR'24

[3] Globally interpretable graph learning via distribution matching. WWW'24

[4] D4Explainer: In-Distribution GNN Explanations via Discrete Denoising Diffusion. NeurIPS'23

[5] Graphon-Explainer: Generating Model-Level Explanations for Graph Neural Networks using Graphons. TMLR'24

[6] MAGE: Model-Level Graph Neural Networks Explanations via Motif-based Graph Generation. ICLR'25

[7] Interpretable Sparsification of Brain Graphs: Better Practices and Effective Designs for Graph Neural Networks. KDD'23

---

> ### Author Response · Authors · 2025-02-23
>
> We sincerely appreciate the time and effort you've dedicated to reviewing and providing invaluable feedback to enhance the quality of this paper. We provide a point-by-point reply below addressing your concerns and questions.
>
> (1) **There exist global-level graph explainers; compare with them**. This is a misunderstanding. All of your cited papers are global-level graph factual explanation methods, i.e., finding significant subpatterns (subgraphs/node features/nodes/edges) that contribute the most to GNN prediction. However, what we mean is that there exists barely any global-level graph "COUNTERFACTUAL" explanation method, which aims to find minimal graph edits such that the graph classification provided by the classifier is altered. Graph factual and counterfactual explanation methods are two distinct types of tasks whose methods cannot be mutually generalized.
>
> (2) **Incorporated additional datasets and baselines from the mentioned baselines**. This is a misunderstanding stemming from point (1). As clarified above, graph factual and counterfactual explanation methods are two distinct types of tasks whose methods cannot be mutually generalized. Hence, the datasets and baselines in those works are not applicable in counterfactual scenarios.
>
> (3) **Identifying common influential motifs is not entirely new**. First, the two cited works are graph factual explanation methods which identify motifs as decisive subgraphs for GNN prediction for one input graph. However, we focus on graph counterfactual explanations (see their difference in point 1). Second, those works are individual-level graph explanation methods, while ours is global-level. Our method finds significant subgraphs based on their frequency of appearance in the input graph set, which leaves their significance to the GNN prediction to the greedy summary of counterfactual subgraph mappings (CSMs) in Section 3.3. This massively saves computation time while maintaining the effectiveness of our method.
>
> (4) **Assertion that comprehensibility is unequivocally "better" than other metrics (e.g., proximity) demands more rigorous justification**: We did not make this claim. We simply highlight that this is an intuitive and naturally important perspective not covered by other proposed metrics. This does not mean that other metrics are less important than ours; they are all important.
>
> (5) **There is no readily available ground-truth motif to benchmark comprehensibility**. We believe this is a misunderstanding; we did not mention ground-truth counterfactuals either in Figure 3 or in the computation of the comprehensibility metric.
>
> (6) **Quality of $G$ strongly influences the comprehensibility**. The reviewer is correct that the quality of set G influences the scores. However, this is true for all existing metrics referred in the paper. Therefore, we do not deem this a issue of our proposed comprehensibility metric.

---

### Review · Reviewer_aPzD · 2025-02-17

**Summary Of Contributions:**

The paper presents GlobalGCE, a novel method for global counterfactual explanation (GCE) of GNNs. The main contribution is the identification of subgraph mapping rules that provide high-coverage, interpretable counterfactual explanations, distinguishing it from existing local-level GCE methods. Also, the authors propose comprehensibility, a metric that quantifies the human interpretability of counterfactual graph edits.

Overall, I find the work interesting and valuable, with contributions to GNN interpretability.

**Audience:**

Yes

**Claims And Evidence:**

Yes

**Requested Changes:**

Overall, I find the work interesting. Here are some suggested revisions:

- Clarify the introduction, especially the discussion on GCFExplainer ("In this paper, we aim to..."), to make the motivation and limitations clearer upfront. I find it hard to follow without reading the following sections.
- Provide a detailed computational complexity analysis of each component (subgraph generation, autoencoder training, greedy summarization) to assess scalability.
- Discuss the generalization of the method to multi-class classification and regression tasks.

**Strengths And Weaknesses:**

**Strengths**

The paper's motivation is clear and the subgraph mapping-based counterfactual generation method is a novel contribution to GNN interpretability.

The use of a variational autoencoder for counterfactual subgraph generation is well-justified and efficiently optimized. And the comprehensibility metric is a valuable addition that enhances the interpretability of counterfactual explanations.

**Weaknesses**

- While the proposed method is scalable compared to local GCEs, the subgraph search and counterfactual generation remain computationally expensive. I would like to see more insights here.

- The method assumes a fixed set of counterfactual subgraph mappings, which may not generalize well to highly heterogeneous graphs.

- The method is evaluated on binary graph classification tasks and may not generalize to multi-class classification or regression problems.

---

> ### Author Response · Authors · 2025-02-24
>
> We appreciate the reviewer aPzD's recognition of clear motivation, novel explainability framework, effective methodology, and valuable new metrics. Here is the point-by-point answers addressing the reviewer's concern:
>
> (1): **Subgraph search and counterfactual generation is expensive.**
> Answer: (a) Subgraph search. We utilize gSpan [1], a mature algorithm for frequent subgraph generation. Higher the minimum appearance rate $\tau$ will reduce the average subgraph sizes $k$ by pruning more branches of the search space, as only frequent subgraphs meeting the threshold are explored. Therefore, although the time complexity can be theoretically high, we can still conduct gSpan when the graph size scales to be very large. Specifically, if gSpan does not return results after a predetermined time period, we increase the minimum appearance rate, and the running time will decrease polynomially/super-polynomially (refer to Fig 5 of [1]). Finally, there will be a significant subgraph set achived within pre-determined acceptable time frame. (b) Counterfactual generation. The time complexity for training the subgraph counterfactual VAE can be analyzed per training iteration. For each significant subgraph $g$, we process all graphs containing it (denoted as $|G_g|$) in batches of size b. For each graph in a batch, we perform VAE operations on the subgraph ($O(n^2)$ for n-node subgraphs), subgraph matching to find insertion locations ($O(Nn)$ for N-node original graphs), insert the counterfactual subgraph ($O(n^2)$), and run forward/backward passes through the GNN ($O(N^2)$). With i training iterations, the total time complexity is $O(i * |G_g|/b * (n^2 + Nn + N^2))$. Generally, as shown in Table 2 in the paper, our method consumes significantly less time than the only existing global-level graph counterfactual explanation work, GCFExplainer.
>
> [1] Yan, Xifeng, and Jiawei Han. "gspan: Graph-based substructure pattern mining." 2002 IEEE International Conference on Data Mining, 2002. Proceedings. IEEE, 2002.
>
> (2) **Fixed set of counterfactual subgraph mappings may not generalize well to highly heterogeneous graphs**:
> (a) The number of subgraph mappings is a hyperparameter that is configurable based on the dataset. We align this number across datasets in Table 1 just for simplicity. According to a user's requirements, one may adjust this number for best performance.
> (b) The paper demonstrates good performance across multiple heterogeneous datasets (AIDS, PROTEINS, ENZYMES, etc.) which already have diverse graph structures.
>
> (3) **The method cannot be generalized to multi-class classification or regression**:
> This is a misunderstanding. In fact, our method can be applied to multi-class classification and regression with minimal modification. Specifically, our method only utilizes the ground-truth binary classifier when computing the counterfactual prediction loss $l(\phi(G^{cf}), y^*)$, i.e., negative log-likelihood between the GNN prediction $\phi(G^{cf})$ and the desired label $y^*$, in (6). This can be simply substituted by cross-entropy loss for multi-class classification and mean squared error loss for regression.

---

### Author Response · Authors · 2024-12-28
**A Sincere Request for Uploading Reviews**

Dear Reviewers,

We submitted manuscript #3558 approximately two months ago and noticed that some reviews are still pending. We understand that you have many commitments and appreciate your valuable time and expertise.

If possible, we would greatly appreciate it if you upload your review. Your insights are important to us in improving our work.

Thank you for your consideration!

Best regards,

Authors of Submission #3558

---

### Author Response · Authors · 2025-02-24
**Manuscript Updated**

Dear Editor and Reviewers,

The manuscript is edited according to the reviewers' requested changes, some of the edits are marked with red fonts. Thank you again for your efforts in reviewing our work!

Sincerely,

Authors from Submission 3558.

---

### Decision · Action_Editor_db2i · 2025-03-22

**Recommendation:** Accept as is

**Comment:**

This paper proposes GlobalGCE, a novel framework for global counterfactual explanations in Graph Neural Networks (GNNs). Instead of editing each graph individually, GlobalGCE identifies a small set of subgraph-mapping rules that can flip predictions of many graphs to a desired class. The method involves three steps: identifying frequently occurring subgraphs, learning subgraph replacements via an autoencoder, and selecting rules to maximize coverage. The authors also introduce a new comprehensibility metric to evaluate interpretability. Experiments show that GlobalGCE achieves higher label-flipping rates and provides more concise and interpretable explanations than existing local-level approaches.

Overall, the approach is interesting, and the reviewers are satisfied with both the paper and the rebuttal. I therefore recommend acceptance.

**Audience:**

The Graph Neural Networks (GNN) is one of the important topics in ML, and many researchers will be interested in this work.

**Claims And Evidence:**

The claims in the submission are well justified by the experiments.